# Developing Trusted Voices for Planetary Health: Findings from a Clinicians for Planetary Health (C4PH) Workshop

Michael Xie [1,*], Vanessa Góes [2,†], Melissa Lem [3], Kristin Raab [4], Tatiana Souza de Camargo [5], Enrique Falceto de Barros [6], Sandeep Maharaj [7] and Teddie Potter [8]

[1] School of Public Health, University of Washington, Seattle, WA 98195, USA
[2] LeBioME-Bioactives, Mitochondria and Placental Metabolism Core, Institute of Nutrition Josué de Castro, Federal University of Rio de Janeiro, Rio de Janeiro 21941-902, Brazil
[3] Department of Family Practice, University of British Columbia, Vancouver, BC V6T 1Z3, Canada
[4] Environmental Health Division, Minnesota Department of Health, St. Paul, MN 55164, USA
[5] Department of Education and Curriculum, Federal University of Rio Grande do Sul, Porto Alegre 90010-150, Brazil
[6] Department of Chemistry of Life and Health, Federal University of Rio Grande do Sul, Porto Alegre 90010-150, Brazil
[7] School of Pharmacy, University of the West Indies, St. Augustine, Trinidad and Tobago
[8] School of Nursing, University of Minnesota, Minneapolis, MN 55455, USA
* Correspondence: mxie1000@gmail.com
† Member of the Nova Network, Baltimore, MD 21231, USA.

**Abstract:** Climate change, biodiversity loss, and other environmental changes are rapidly impacting the health of people worldwide, but many clinicians and other health professionals feel unprepared to deal with this burgeoning issue. During the Planetary Health Annual Meeting held in Boston in late 2022, the Clinicians for Planetary Health (C4PH) working group hosted a workshop that highlighted the latest findings of clinicians' attitudes towards climate change, connections with the related fields of lifestyle medicine and integrative health, lessons learned from implementing "one minute for the planet" in a rural Brazilian clinic, and the benefits of clinicians prescribing time in nature for their patients. This article ends with a few suggestions for healthcare providers to begin implementing planetary health into their professional practice.

**Keywords:** planetary health; healthcare professionals; climate change; sustainable healthcare; wellbeing

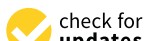



## 1. Introduction

Human health is inextricably linked to the health of the biosphere [1]. Yet, human activity continues to disrupt Earth's natural systems by emitting greenhouse gases and degrading land and marine ecosystems through natural resource overexploitation, agriculture, and pollution, among other activities [2–5]. These actions are causing drastic and catastrophic changes to our air, water, weather, and ecosystems that are affecting all life on Earth [6]. Instead of caring for the earth's natural systems, humans have gone through an unprecedented time of destruction [7].

Planetary health is a concept that connects human-caused disruptions of natural processes on Earth to negative consequences within human health and encourages interdisciplinary collaboration [8]. From the increased frequency of wildfires impacting lung and cardiovascular health to the disruption of food systems to the exacerbation of mental health issues due to trauma after natural disasters, the impact of humans' disruption on the planet and on human health is now impossible to ignore [9–11]. Other examples include heat-related illnesses from extreme heat events, zoonotic and vector-borne diseases from changing climatic conditions, and injuries and deaths related to severe weather events [12–15]. Between 2030 and 2050, it is estimated that climate change will cause an additional 250,000 deaths

per year [16]. Planetary health encompasses more than just climate change; rather, it includes all forms of environmental degradation caused by humans and emphasizes the manifold connections between individual and global well-being [17]. For example, the biodiversity crisis is furthermore linked to a rapid disappearance of pollinators, threatening global food security, and witnessing the overall degradation of the earth has led many to develop mental health problems [18,19].

There is still time to prevent the worst consequences of humans' negative impact on the Earth [20]. Mitigating these global environmental changes can have the co-benefits of improving human health and improving the large-scale systems on which all life on this planet depends [21,22]. Health professionals see the health impacts of natural degradation first-hand in their patients and can directly connect concerning phenomena in natural life support systems to individual health issues.

Healthcare professionals are also trusted voices—they have qualities that make them trustworthy: technical competence (they are the experts in healthcare) and moral character (they are in the medical profession to heal others) [23–25]. This gives healthcare providers a lot of power to plant seeds: that planetary degradation and climate change are happening locally and patients can take actions to stay safe and healthy. Others have also exhorted clinicians to become involved in the health–environment nexus [26–28].

At the Planetary Health Annual Meeting in November 2022, the Clinicians for Planetary Health (C4PH) group, an initiative of the Planetary Health Alliance (PHA), hosted an event titled "Developing Trusted Voices for Planetary Health: A C4PH Workshop." The workshop explored how all clinicians need to be prepared to discuss planetary health in their practice, the current state of clinician knowledge on planetary health and the barriers to implementation, and effective strategies to communicate planetary health challenges and solutions.

## 2. Summary of Event

The event was begun by Kristin Raab, Director of the Climate & Health Program at the Minnesota Department of Health, who presented findings from a survey of healthcare providers performed in 2021 in Minnesota, a state in the United States, which found that 75% ($n$ = 3303) of respondents agreed that climate change is happening, and 60% ($n$ = 2696) agreed that "climate change directly or indirectly impacts the health of my patients, clients, or community members" [29]. Among those who agreed or did not disagree that climate change impacts health (78% of the respondents), 76% of respondents expressed concern about the health impacts of climate change on their patients [29]. Almost 60% indicated that healthcare professionals are positioned to help patients understand the health impacts of climate change and that they should have an active role in discussing it with their patients, but only 1 in 5 felt comfortable or prepared to discuss the health impacts of climate change with their patients [29].

Healthcare providers are noticing the health impacts of climate change in their patients, and they are concerned and think they should help their patients, but they do not feel comfortable or prepared to have a conversation. Only 4% of survey respondents reported that they discuss the impacts of climate change on health with most of their patients, and the vast majority (83%) stated that they never discuss the health impacts of climate change or only with a few patients [29].

The number one barrier to conversing about climate change was a lack of knowledge on how to approach the issue [29]. The survey respondents provided a number of suggestions for resources that they thought would be helpful to having climate and health conversations [29]. The top responses included continuing education courses, research articles, and patient education materials [29].

Vanessa Goes MSc, a Brazilian integrative health professional and Ph.D. candidate in food science, gave a thorough introduction to lifestyle medicine (LM), in which she relayed the interdependency between environmental conditions and an individual's lifestyle: the environment affects us, and vice versa [30,31]. Not only that, many medical experts

outside of lifestyle medicine agree that the choices and habits we make in our daily lives play a significant role in determining human health as well [32]. Behavior (definitive actions, such as smoking/not smoking) and lifestyle (which comprises a wide range of behaviors based on values, cultural context, and attitudes) are leading forces behind the Anthropocene-era emergence of the increasing burden of non-communicable diseases [33]. With this in mind, integrating a lifestyle medicine approach, such as health coaching, to conventional medical practice is a starting point for improving overall health, both emotional and physical well-being, by increasing patient engagement as well as enhancing motivation and readiness for change. Commonly used approaches to support behavioral change within lifestyle medicine include positive psychology, which focuses on positive subjective experiences and emphasizes interventions to boost gratitude, hope, creativity, courage, perseverance, a sense of purpose, and social connectivity, as well as motivational interviewing, which uses a collaborative conversational style to deal with ambivalence and help the patient develop a deeper ability to self-reflect [34,35]. All of this lies within the transtheoretical model of change, which can be used to understand the stages individuals progress through to achieve a change in health behaviors, from precontemplation to long-term maintenance, with the help of concrete goal-setting and the measurement of success each step of the way [36]. Furthermore, lifestyle medicine interventions are effective in helping patients understand the links between a healthy lifestyle and environment, and LM providers are uniquely positioned to coach patients toward climate-healthy behavior changes that heal both people and the planet. Behavior changes can take place within the "pillars" of nutrition, exercise, avoidance of risky substances, stress management, sleep, and healthy relationships and connections [37]. For example, eating a plant-forward diet can not only help improve individual human health but also reduce the impacts of climate change moving forward [38]. Besides these tangible co-benefits, LM values include a comprehensive health perspective and interdisciplinary collaboration, which are all aligned with the field of planetary health [39]. As such, according to Goes, the LM framework is ideal for integrating planetary health into the everyday practice of medicine [40–43].

Dr. Enrique Falceto de Barros, a family doctor in a rural community in Brazil, presented a clinical case of a mother with dyspnea and wheezing and her child with fever and rash in the summer. It turned out that the baby had dengue—the first diagnosis in this rural, mountainous community that is usually cooler than the rest of Brazil. The mother was diagnosed with long COVID, and her asthma treatment was updated to be less polluting and better for the patient's health. From his many years of practice, primary care providers are on the frontline of implementing planetary health solutions as part of a broader framework of reaching net zero in the healthcare sector. This can come as simply as a switch in a prescription, as Dr. Barros has shown. Dr. Barros also introduced his concept of "one minute for the planet", where he dedicates a short amount of time embedded within each clinic visit to advise each patient on what they can do to improve their health and reduce their impact on the planet [44]. In the previous example, Dr. Barros explained how some inhalers produce much fewer emissions than others and that our individual actions can make a difference in preventing the worsening of air pollution (which can trigger her asthma) in the future, such as by eating less red meat [45,46]. One of the largest causes of deforestation in Brazil specifically, and of the subsequent burning of forests, is the expansion of pasture for beef cattle grazing [47]. There is no need to force a discussion of environmental factors; rather, they naturally come up in the course of the visit. Dr. Barros has shown that clinicians can talk to patients about climate change without sacrificing patient care in a short amount of time. Connecting the dots between environmental influences on health and the current symptoms for patients can lead to a greater understanding of their health and an increased sense of agency to positively impact both themselves and their communities.

Dr. Roberto de Almeida, director of the Ideia Ambiental Institute, an NGO dedicated to promoting planetary health in Brazil, started off with the premise that how we practice medicine today is unsustainable, given how the planet is warming at an unprecedented

rate, and the fact that healthcare is among the most polluting service sectors [48,49]. With this emergency, how should we inform the next generation and train them on how to take action? Using the theme of "Healthy People, Healthy Planet" from the 2016 American College of Lifestyle Medicine national conference as a foundation, Dr. Almeida sought to design an integrative health program at his school based on the three pillars of self, society/community, and environment/ecosystem. Almeida also proposed that the Anthropocene we are currently living in is an age of disconnections: individuals are disconnected from society, and humans at-large seem to be disconnected from the natural world around us. To remedy this, integrative health aims to reconnect all three aforementioned pillars. He further urged the audience to become more mindful of restoring harmony and taking care of the planet in our everyday actions, including in healthcare. Dr. Almeida concluded with a real-life example of his own work in the Paraná watershed, where he and his colleagues created new teaching activities, including 55 seminars for over 4000 local educators, throughout the pandemic, in order to instill a new type of "planetary intelligence".

Dr. Melissa Lem, founder and director of PaRx and president of the Canadian Association of Physicians for the Environment (CAPE), wrapped up the case studies of health professional action with a description of the patient and planetary health benefits of nature prescribing. PaRx, a prescription (Rx) for parks, is Canada's national nature prescription program that aims to help clinicians prescribe time in nature for their patients in a simple, fun, and effective manner [50]. She described the many research-proven benefits of being near nature, including improved hypertension, heart disease, and diabetes; reduced anxiety and depression; better academic performance; and symptoms of attention deficit hyperactivity disorder (ADHD) in children [51–57]. Dr. Lem also discussed the significant health and infrastructure benefits of increased tree cover and green space in urban settings and their inequity of distribution, highlighting the fact that the majority of hospitalizations in the Vancouver Coastal Health Authority during the 2021 heat dome came from the Downtown Eastside of Vancouver, one of the most economically and nature-deprived areas of the city [58,59]. Elsewhere, too, traditionally marginalized and resource-poor communities have less access to natural spaces compared to wealthier areas [60,61]. Turning to PaRx, she reviewed the evidence behind its recommendation that people spend at least two hours in nature each week, for at least 20 min each time [62]. She also described the international reach of its message, PaRx having been recognized by the World Health Organization at COP26 in its Special Report on Climate Change and Health as an effective way to inspire the protection and restoration of nature as the foundation of our health [63]. Celebrating the power of cross-sectoral action, she outlined the overwhelming international media attention resulting from PaRx's collaboration with Parks Canada to prescribe free national park passes [64]. She also highlighted research indicating that connecting people to nature increases their engagement in pro-environmental behaviors to broader climate action and supports a deeper sense of connection with the natural world and overall well-being [65–68].

## 3. Discussion

Even though our speakers may have come from a wide range of backgrounds, there were many meaningful cross-connections between their separate talks. For example, the survey in Minnesota demonstrated that clinicians understand their role in discussing climate change, yet do not feel comfortable doing so due to a lack of time and knowledge. Two examples from South America, Dr. Barros' "one minute for the planet" and Dr. Almeida's theoretical knowledge-building platform, demonstrate simple ways that have already been executed in a real-life teaching setting and can serve as a foundation for more clinically specific educational materials. Furthermore, the concept of "one minute for the planet" shows how to turn this widespread motivation to take action into reality without sacrificing any necessary clinical care or feeling the need to insert something additional. In many cases, as Dr. Barros has highlighted, explaining climate change is part of explaining the causes of the present illness as well as the next steps patients can take. Not only are

clinicians eager to get involved, but patients are also receptive to hearing climate-related messages, especially if they directly apply to their own health. As such, "one minute for the planet" also addresses the next largest barrier cited in the Minnesota study (i.e., the concern that patients would not be interested).

The talk on lifestyle medicine connects seamlessly with Dr. Almeida's educational framework and Dr. Lem's introduction of the PaRx system, which is a concrete example of evidence-based lifestyle medicine. By taking a lower-cost preventative approach of encouraging individuals to spend more time in nature to boost physical and mental well-being instead of waiting to treat diseases after they have already occurred, PaRx embodies the core principles of lifestyle medicine, such as taking a comprehensive perspective on health and using exercise as a "prescription." PaRx and lifestyle medicine not only relate to each other well but also to the separate concept of planetary health, as both similarly emphasize the interconnectedness of all life forms on this planet, including in the natural world [69]. PaRx is a prime example of the "reconnection" Dr. Almeida explained between individuals and the environment that must take place in this era of disconnection. This reconnection can foster a new sense of planetary intelligence that can then further spur other pro-climate actions in patients' daily lives—healthy behaviors that benefit the Earth as well. Such an effort ties back to millennia-old wisdom from many Indigenous communities around the world regarding the importance of protecting our environment and building a cooperative, not exploitative, relationship with the planet [70]. Furthermore, focusing on prevention and promoting long-term well-being instead of hospitalization is a core aspect of creating a low-carbon healthcare system in countries of all economic classifications, thereby reducing the impact of humans on our environment and helping mitigate the worsening of climate change [71,72].

Expanding beyond the scope of only this event, the work of these various speakers connects to the larger scholarship of planetary health in many ways. One similarity to the existing literature is how the speakers have advocated for a bigger-picture approach to planetary health via systems thinking, which emphasizes "a holistic approach for understanding the dynamic interactions among complex economic, environmental, and social systems and for evaluating the potential consequences of interventions" [73]. This approach was heavily emphasized in both Goes' and Almeida's presentations through their cogent analyses of reconnecting humans to a planet-wide web of life and has been tied to planetary health by other researchers previously as a way to improve the understanding of the complex relationships between human health and the natural world [74,75]. By bringing together research, theoretical frameworks, and multiple examples of implementing planetary health in clinical settings worldwide, this event has filled a crucial gap in the literature: that of bridging the divide between what we already know and how to act upon that knowledge on a day-to-day basis. However, this event inherently could not capture the entire depth and breadth of planetary health, given its limited time frame. For instance, much has been written about education efforts for future clinicians, exact roles for different types of clinicians (e.g., nurses), interdisciplinary scientific research that goes beyond suggestions for clinicians (e.g., clinical ecology), or Indigenous perspectives, which the speakers did not cover [76–80]. In the future, researchers can continue to find new connections between these disparate aspects of planetary health and investigate further how more clinicians can use their trusted voices to advocate for even greater changes in climate and other environmental policies (e.g., regarding biodiversity, deforestation, etc.) at all levels, and not restrict planetary health to only clinical spaces.

## 4. Conclusions

This C4PH event laid out the current state of clinicians' knowledge of climate change and other human-environmental changes (e.g., loss of connection with nature) and proposed a wide range of solutions, including "one minute for the planet", adopting approaches from lifestyle medicine, a comprehensive planetary health education framework at the undergraduate level, and prescribing nature as an evidence-based treatment. Com-

bining these strands of wisdom together, clinicians from any setting can take away some concrete tips.

First, any mention of planetary health in a clinical setting does not have to be time-consuming: even a sentence explaining the ties between climate change to the symptoms or ailment at hand or elaborating upon nature-based solutions is a good start. Second, planetary health does not have to be forced into clinical interactions: as PaRx and the field of lifestyle medicine have shown, there are many co-benefits of actions that are healthy both for humans and the planet (e.g., eating less red meat, being in nature). Many patients are already worried about climate change and similar environmental issues, and discussing this topic explicitly can spread awareness about the environment–health nexus and empower patients to make a positive impact, both for themselves and the world around them [81,82].

Finally, if all of this seems overwhelming, try seeking out like-minded professionals. An increasing number of institutions, such as the Center for the Environment and Health at Massachusetts General Hospital in the US or Sunway Centre for Planetary Health in Malaysia, are creating new offices dedicated towards the environment, so check with your workplace as well if you work in such a setting [83,84]. Regardless of where you work, see if your colleagues are interested and team up with them to pursue further action. Many organizations, such as Health Care Without Harm, the Environment Group of the World Organization of Family Doctors (WONCA), or Grupo de Estudos em Saúde Planetária in Brazil, provide additional sources of wide-reaching community or tools for health professionals of all kinds to implement planetary health that provide solutions for various types of healthcare settings [85–87]. The Planetary Health Alliance (PHA) houses regional hubs to ensure any planetary health solution is context-specific and uses a global social networking platform called Hylo [88,89]. Other climate-health organizations are being created and expanding all around the world, so a community is likely not far from you. (See Figure 1 for a condensed summary of potential actions.) With these tools in mind, health professionals around the world should be able to tackle the complex issue of planetary health, one patient or community at a time.

---

**Key tips:**

- **Pressed for time? Start off with just a sentence.**
- **Focus on lifestyle choices that present health and climate co-benefits simultaneously.**
- **Seek out like-minded colleagues and friends.**

---

**Figure 1.** Key tips for clinicians to implement planetary health into their professional practice.

**Author Contributions:** Conceptualization, M.X.; writing M.X., V.G., M.L. and K.R.; editing M.X., V.G., K.R., T.S.d.C., E.F.d.B., S.M. and T.P. All authors have read and agreed to the published version of the manuscript.

**Funding:** This research received no external funding.

**Data Availability Statement:** No data were used in the publication of this article.

**Conflicts of Interest:** The authors declare no conflict of interest.

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
