# Peer review of "Developing Trusted Voices for Planetary Health: Findings from a Clinicians for Planetary Health (C4PH) Workshop"

_challenges, doi:10.3390/challe14010017_

Round 1

Reviewer 1 Report

Peer-review of ' Developing Trusted Voices for Planetary Health: Findings from a Clinicians for Planetary Health (C4PH) Workshop’ (in Challenges, manuscript # 2223572)

General Comments

The authors have crafted an elegant, thought-provoking manuscript that is worthy of publication and widespread dissemination. In the emerging concept of planetary health, the role of healthcare providers is often overlooked in solutions-based discourse. The manuscript builds upon previously published papers, but this submission is unique in the way it sews together case presentations and real-world programs from experts with disparate but overlapping experiences. The manuscript provides helpful advice for clinicians who might not otherwise appreciate their potential role, or how they can engage with patients/clients.

Overall, this manuscript is close to qualifying for publication. With minor revisions and some tightening, this could be a notable paper, one that has the potential to encourage wider conversations.

Suggested Revisions

1.      Please add at least a few lines early on in the manuscript with a reference or two defining planetary health. What is it, and what does the term mean? Although there is one sentence (lines 54 to 57) underscoring that biodiversity losses are distinct from climate change, the general theme of the manuscript seems to infer that planetary health is all about climate change. As the authors know, it is far more than that. Placing greater emphasis on the union of personal health (the clinician) and public health (the places) might allow clinicians to see the relevancy of planetary health in their everyday work. See

Ezell JM, Griswold D, Chase EC, Carver E. The blueprint of disaster: COVID-19, the Flint water crisis, and unequal ecological impacts. The Lancet Planetary Health. 2021 May 1;5(5):e309-15.

Schmidt L, Mialon M, Kearns C, Crosbie E. Transnational corporations, obesity and planetary health. The Lancet Planetary Health. 2020 Jul 1;4(7):e266-7.

Robinson JM, Redvers N, Camargo A, Bosch CA, Breed MF, Brenner LA, Carney MA, Chauhan A, Dasari M, Dietz LG, Friedman M. Twenty important research questions in microbial exposure and social equity. Msystems. 2022 Feb 22;7(1):e01240-21.

2.      The manuscript doesn’t really capture the many conversations that have previously discussed the role of clinicians in planetary health. See these papers and please add at least some of them to reinforce the argument. It is understood the current authors are providing a prescriptive approach, but that Rx will be strengthened by contextual discourse.

 Xie E, de Barros EF, Abelsohn A, Stein AT, Haines A. Challenges and opportunities in planetary health for primary care providers. The Lancet Planetary health. 2018 May 1;2(5):e185-7.

Veidis EM, Myers SS, Almada AA, Golden CD. A call for clinicians to act on planetary health. The Lancet. 2019 May 18;393(10185):2021.

Capon AG, Talley AC NJ, Horton RC. Planetary health: what is it and what should doctors do?. Medical Journal of Australia. 2018 Apr;208(7):296-7.

3.      Lines 135-136 state that ‘ according to Goes, the LM framework is ideal for integrating planetary health into the everyday practice of medicine.’. This is an important and valid statement. However, several authors have argued this very point, and from a research ethics point of view, at least some of that work should be cited. See

Pathak N, Pollard KJ, McKinney A. Lifestyle Medicine Interventions for Personal and Planetary Health: The Urgent Need for Action. American Journal of Lifestyle Medicine. 2022 Sep;16(5):589-93.

Pathak N, McKinney A. Planetary health, climate change, and lifestyle medicine: threats and opportunities. American Journal of Lifestyle Medicine. 2021 Sep;15(5):541-52.

Pathak N, Pollard KJ. Lifestyle medicine prescriptions for personal and planetary health. The Journal of Climate Change and Health. 2021 Oct 1;4:100077.

Logan AC, Prescott SL, Katz DL. Golden age of medicine 2.0: lifestyle medicine and planetary health prioritized. Journal of Lifestyle Medicine. 2019 Jul;9(2):75.

4.      Lines 177-178 describes Dr. Melissa Lem as the founder and director of PaRx. Please clarify what PaRx stands for and provide a sentence or two about the mission/values of PaRx. Lines 180 to 187 provide quite a few statements presented as fact, and in a scientific publication will need to be references. There are lots of references to show that there are relationships between nature (e.g., residential proximity to greenspace) and health outcomes, but as written it states that connecting people to nature (that is, an active intervention) leads to multiple research-proven outcomes. Please add some references to support the intervention claims. I would also recommend adding a few references on nature connectivity or nature relatedness as distinct from whether or not a person has access to nearby greenspace. What are the equity concerns with nature access? Clinicians may want to know more about the concerns of their patients/clients in this regard.

Tomasso LP, Cedeño Laurent JG, Chen JT, Spengler JD. Implications of disparities in social and built environment antecedents to adult nature engagement. Plos one. 2022 Sep 23;17(9):e0274948.

Reviewer 2 Report

The report brings an important approach to a seed intervention by Health professionals regarding Planetary Health and its relation to human health but  in individual basis.

in my view, this is important but not sufficient for the    Necessary changes society and economy need to undertake for mitigating climate change Health impacts. So, in the discussion it would be relevant to highlight the need of global public polities to curb deforestation and biodiversity loss on top of these small interventions.

Author Response

Please see attachment for point-by-point response to your feedback. Thank you! 

Reviewer 3 Report

Thanks for this good summary of the C4PH event. Overall, this is a pretty straightforward description of the event and distillation of lessons learned, so I have no comment or suggestion to make. I also appreciate the attempt to connect the different case studies presented during the event.

There is only one question that I have - on Page 3, Line 113, it says "Two leading forces behind the 112 increasing burden of non-communicable diseases are lifestyle and behavior" - aren't lifestyle and behavior similar or even synonymous?

Author Response

Update: please see attachment for how we've improved the manuscript based on your feedback. Thank you for taking the time to thoughtfully read what we have! 

Reviewer 4 Report

Dear Authors, 

I read your manuscript carefully. Please find below my comments.

1) Introduction is very long. It should be more concise. It should give an overall overview of the topic, but not details that are not fundamental for the understanding of the other sections of the manuscript.

2) Description of the conference is adequate

3) Discussion must be improved. Currently, it is only a focus on presentations' content. It should connect the "Summary of Event" to the current literature, discussing the similarities, the gaps, and drawing relevant links.

4) In the conclusions, lines 254-263 seems and "advertisement" of the PHA. I think this is not suitable for a scientific publication. Despite PHA has an honorable and noble objective (and I admire this initiative), these sentences should be rewritten.

Author Response

(The authors gave the same response as above.)
